# Teenage Blues: Predictors of depression among adolescents in Nigeria

**Adefunke DadeMatthews**[1,2]*, **Chukwuemeka Nzeakah**[2,3]©, **Lucky Onofa**[2]©, **Oluwagbemiga DadeMatthews**[4‡], **Temitope Ogundare**[5‡]

1 College of Human Sciences, Human Development and Family Studies, Auburn University, Auburn, Alabama, United States of America, 2 Department of Clinical Services, Neuropsychiatric Hospital, Aro, Abeokuta, Nigeria, 3 Kent & Medway All Age Eating Disorder Service, North East London NHS Foundation Trust, Maidstone, Kent, United Kingdom, 4 School of Kinesiology, College of Human Sciences and Education, Louisiana State University, Baton Rouge, Louisiana, United States of America, 5 Department of Psychiatry, Boston Medical Center, Boston, Massachusetts, United States of America

© These authors contributed equally to this work.
‡ OD and TO also contributed equally to this work.
* aod0006@auburn.edu

**Data Availability Statement:** All relevant data are within the paper and its Supporting information files.

**Funding:** The author(s) received no specific funding for this work.

## Abstract

### Background

Depressive disorders, with a prevalence of 15–21%, are among the most common disorders in children and adolescents, and increases the risk of suicide, the second leading cause of death in children aged 10 to 19.

### Aim

To determine the prevalence and correlates of depressive disorders among senior students attending secondary schools in Abeokuta.

### Method

The study was conducted in five schools randomly selected from a representative sample and was carried out in 2 phases. In the first phase, students were selected via systematic random sampling and given consent forms and GHQ-12 to administer to the parents. In the second phase, students who returned a signed informed consent form and filled out GHQ-12 were interviewed using MINI-KID, Rosenberg's Self-Esteem Scale, Family-APGAR, and sociodemographic questionnaire. Multivariate regression analyses were conducted with p-value <0.05 as level of significance.

### Results

The mean age was 15.3 years (SD = 1.27); 48.8% were male. The twelve-month prevalence of major depression was 11.3% and dysthymia was 1.4%. In the final regression analysis, female gender [OR = 4.3, p = 0.046], the experience of bullying [OR = 7.96, p = 0.004], difficulty getting along with friends, [OR = 7.5, p = 0.004], history of sexual abuse [OR = 8.1, p = 0.01], and perceived family dysfunction [OR = 4.9, p = 0,023] were found to be independent predictors of depressive disorders.

**Competing interests:** The author(s) received no funding for this work.

## Conclusion

Depressive syndromes are a significant health burden in adolescents. Being female, being bullied, having a history of sexual abuse, and family dysfunctionality are risk factors associated with depression among these population.

## Introduction

Depressive disorders are among the most common disorders in children and adolescents, with high prevalence of 12% [1] and rising to 21% [1–4]. Depression increases the risk of suicide, the second leading cause of death in children and young adults between ages ten and twenty four [5] and one of the top ten leading causes of death across all ages [5, 6]. The reported prevalence of depression among the adolescent population varies widely due to differences in the methodology of various epidemiological studies. Current literature suggests that major depressive disorder is more prevalent than dysthymia and other subtypes of depression [7, 8]. Before puberty, there is no significant difference in the prevalence of depression between the genders, but after that, an increased incidence in females is widely documented across various cultures [9, 10]. Overall, the prevalence of depression rises with increasing adolescent age [11–13].

Etiological models suggest that exposure to multiple antecedents interacts with innate characteristics to increase adolescents' risk of depression. Individual risks for depression in adolescents include certain genetic factors, female gender, endocrine dysfunction, negative cognitive styles, sub-clinical depressive symptoms, specific personality traits, and problems in self-regulation/coping behaviors, while risks from the external environment include faulty parent-child relationships, adverse life events and on-going interpersonal difficulties [14–17].

The genetic contribution to the etiology of adolescent depression appears to be moderate, with heritability estimates ranging between 40–70% [14, 18]. There is a three to four-fold increased risk of depression in the offspring of adults with unipolar depression compared to children of non-depressed parents [19], which is significantly higher in cases of post-natal maternal depression [20]. This elevated risk is also seen in other forms of parental psychopathology, although the association's strength is greatest for parental depression.

Studies have shown that low self-esteem, characterized by global self-devaluation, perceived incompetence, and negative attributional styles, are cognitive factors that markedly increase the risk of depression in adolescents [21, 22]. Furthermore, adolescents with a highly emotional temperamental style (neuroticism) characterized by reacting quickly to everyday events, being easily brought to tears, or easily soothed are also recognized to have a significantly elevated risk of depression [23].

Relational conflicts within the family environment play a major role in the etiology of adolescent depression. Studies show that insecure attachment and parenting characterized by coldness, rejection, harsh discipline, and unsupportive behavior are associated with adolescent depressive symptoms [17]. Parental psychopathology, particularly maternal depression, may contribute to chronic interpersonal stress in the family, compromising the quality of parenting, which may negatively affect youths' psychosocial functioning [24]. Other family-based pathogenic factors include physical abuse, neglect, absent monitoring, marital discord, low family cohesion, lack of authoritative parenting, severe acute disruptions such as sudden death or serious illness in a close relative and sudden parental separation [25–27].

Parental depressive symptoms, perception of poor family functioning, peer problems, low self-esteem, female gender, and large family size are some of the factors that have been

associated with clinically significant depressive symptoms in adolescents [21]. Other factors that are significantly associated with depression in adolescents include alcohol consumption, drug abuse [28], sexual activity [28, 29], and physical violence [28–30]. Lower levels of physical activity has also been linked with severe depressive symptoms [28, 31], while moderate physical activity was linked with reduced risk of depressive symptoms [28].

Depression is markedly increased due to multiple adverse experiences involving longstanding family and more recent friendship events and peer difficulties. Adolescents with poor friendships, characterized by low numbers of friends, infrequent contact, and no intimate relations, are more likely to develop depression, deviant behaviors, and increased social isolation from the desired peer network [22, 32, 33]. Studies have also shown that adolescents who are bullied and those who are bullies are at an increased risk of depression and suicide [34, 35].

Local studies on the prevalence and correlates of depression show findings largely comparable with results from elsewhere. A cross-sectional survey of adolescents in southwest Nigeria found that 5.1% met the criteria for a major depressive disorder (MDD) [4]. A study using the Beck's Depressive Inventory found 9% of school-attending adolescents in another southwestern town in Nigeria to have clinically significant depressive symptoms [21] with a diagnosis of MDD established in 6.9% of the total sample [36]. In another cohort of high school students in a major city in north-eastern Nigeria, a 12% prevalence of depression was reported with 50% of the students who used substances reporting depression [2].

Several studies worldwide have sought to establish the prevalence and associated socioeconomic burden of depressive syndromes in adolescents. However, only a few studies have been done in subsaharan Africa to address the subject [2, 7, 36–39]. This study aims to determine the prevalence and correlates of depressive disorders among senior students attending secondary schools in Abeokuta.

## Materials and methods

The study was carried out in Abeokuta, southwestern Nigeria. It is part of a larger study on anxiety and depressive disorders conducted among secondary school students aged 12–18 years, and the methodology has been described elsewhere [40]. The study was conducted in 2 phases. In the first phase, students were selected via systematic random sampling and given consent forms and GHQ-12 to administer to the parents. In the second phase, students who returned a signed informed consent form and filled out GHQ-12 were interviewed using MINI-KID, Rosenberg's Self-Esteem Scale, Family-APGAR, and sociodemographic questionnaire. Five schools were randomly selected, and from each school, a proportional sampling method, accounting for the sizes of each school, was employed. The sample size was calculated using the formula for estimating proportions [41].

### Study instruments

**Sociodemographic questionnaire.**   This was designed to collect data on the sociodemographic characteristics of the participants and factors linked to depression, such as trauma exposure, history of medical illness, etc. It also collected information about family structure and health-related behaviors.

**Rosenberg's Self-Esteem Scale.**   The 10-item Rosenberg's Self-Esteem Scale [42] measures both positive and negative thoughts about oneself to evaluate participants' overall sense of self-worth. The responses to each question are given on a 4-point Likert scale, with the options being 'strongly agree' to 'strongly disagree.' Higher ratings on the scale correspond to higher levels of self-esteem. It has been tested on samples of teenage girls from Nigeria [21]. In this study, it was utilized to gauge the degree of adolescent self-esteem.

**MINI-KID.** The Mini International Neuropsychiatric Interview version for children (MINI-KID) is a diagnostic interview specifically developed for children and adolescents aged 6–17 years [43]. It was created to give clinicians a quick, valid, and accurate way to diagnose current DSM-IV and ICD-10 psychiatric illnesses and suicidality in child and adolescent populations. It generally has satisfactory psychometric properties, with excellent reliability estimates: kappa values of 1.00 and 0.72 for inter-rater and test-retest reliability [44]. The MINI-KID was utilized to identify depressive disorders in the study sample. The MINI-KID's current timeframe for the specific depressive disorders was adjusted for this investigation to the previous 12 months. It has been used in studies in Nigeria with good psychometric properties [45, 46].

**Family APGAR.** The Family APGAR [47] is a brief screening questionnaire created to get a respondent's opinion of how well their family is doing. It consists of five questions that measure how satisfied respondents are with each of the five aspects of family functioning: adaptability, partnership, growth, affection, and resolve. Each parameter is rated on a 3-point scale: 0 for rarely, 1 for occasionally, and 2 for almost always. A family with a total score of 0 to 3 is likely to be very dysfunctional, 4–6 is likely to be moderately dysfunctional, and 7 to 10 is likely to be highly functioning. In various local investigations, the Family APGAR Score is valid and accurate for evaluating family functioning [48, 49]. The participating students finished it without any assistance.

### Data analysis

IBM SPSS statistics version 25.0 was used to analyze the study's data. The independent t-test and analysis of variance (ANOVA) for continuous variables and chi-square statistics for categorical variables were used to evaluate the relationships between diagnostic categories and various sociodemographic, family, and psychosocial variables. Post-hoc analysis was conducted on statistically significant variables in the ANOVA analyses, and Fisher's exact test was utilized as needed. Statistically significant variables in the bivariate analyses were entered into a multiple regression analysis model to identify independent predictors of depression. The Kolmogorov-Smirnov test was used to determine whether continuous variables such as age, FAPGAR score, overall academic exam score, and self-esteem score were normal.

### Ethical approval

Ethical approval was obtained from the Ethical Committee of the Neuropsychiatric Hospital and the Ogun State Ministry of Education, Science, and Technology, and the administrators of the selected schools granted permission. The parents/guardians of all the participating students also provided written and signed informed consent forms. Additionally, it was made clear to the students that participation was voluntary, and they could decide to withdraw from the study at any time, and not participating would not affect their academic performance. Teachers were not allowed in the room during the interviews to provide extra protection.

### Results

A total of 225 students were selected to participate in the study, 5 (1.96%) of them declined to participate (they were all senior secondary school class 3 students; SSS 3 students) who had a significant test the next day, and 6 (2.35%) were excluded because they were older 18 years. The final sample size was 213.

## Sociodemographic characteristics

Table 1 summarizes the sociodemographic characteristics. The mean age was 15.3 years (SD = 1.27); 48.8% were male. Only 0.9% of fathers and 1.4% of mothers reported no formal education; and 3.8% of fathers were unemployed compared to 6.1% of mothers.

**Table 1. Sociodemographic characteristics of participants.**

| Characteristic | Frequency (n) | Percentage (%) |
|---|---|---|
| **Age Group** | | |
| 12–15 | 119 | 55.9 |
| 16–18 | 94 | 44.1 |
| **Gender** | | |
| Female | 109 | 51.2 |
| Male | 104 | 48.8 |
| **Class** | | |
| SS1 | 54 | 25.4 |
| SS2 | 89 | 41.8 |
| SS3 | 70 | 32.9 |
| **Religion** | | |
| Islam | 69 | 32.4 |
| Christian | 144 | 67.6 |
| **Tribe** | | |
| Yoruba | 189 | 88.7 |
| Igbo | 12 | 5.6 |
| Hausa/Fulani | 3 | 1.4 |
| Others | 9 | 4.2 |
| **Parental Level of Education** | | |
| Father | | |
| No formal education | 2 | 0.9 |
| Primary Education | 44 | 20.7 |
| Secondary education | 54 | 25.4 |
| Tertiary education | 95 | 44.6 |
| Unknown | 18 | 8.5 |
| Mother | | |
| No education | 3 | 1.4 |
| Primary education | 61 | 28.6 |
| Secondary education | 67 | 31.5 |
| Tertiary education | 71 | 33.3 |
| Unknown | 11 | 5.2 |
| **Parental Employment** | | |
| **Father** | | |
| Not Applicable/unknown | 9 | 4.2 |
| Unemployed | 8 | 3.8 |
| Employed | 196 | 92.0 |
| **Mother** | | |
| Unemployed | 13 | 6.1 |
| Employed | 199 | 93.4 |
| Not reported | 1 | 0.5 |

## Child–related psychosocial factors

Table 2 shows child-related psychosocial factors. Sixty-three (29.6%) respondents experienced the loss of a close relative during the previous year; of these, 32 (50.8%) reported a close relationship with the deceased relative. Only 9.9% reported no friends; 13 (6.1%) people admitted to participating in bullying of others; 16.9% reported having experienced bullying; 17 (8.0%) participants had experienced sexual abuse (8.3% of girls vs 7.7% boys); 16 (7.5%) people reported current use of psychoactive substances, with alcohol being the most popular drug (3.3%).

**Table 2. Child-related psychosocial variables.**

| Psychosocial characteristic | | Frequency (n) | Percentage (%) |
|---|---|---|---|
| **Recent Loss** | No | 150 | 70.4 |
| | Yes | 63 | 29.6 |
| **Close Friends** | None | 21 | 9.9 |
| | One | 35 | 16.4 |
| | Two | 55 | 25.8 |
| | Three or more | 102 | 47.9 |
| **Best Friend** | No | 46 | 21.6 |
| | Yes | 167 | 78.4 |
| **Friend Interaction** | Rarely | 9 | 4.2 |
| | Some days | 58 | 27.2 |
| | Everyday | 146 | 68.5 |
| **Getting Along With Friends** | Often quarrel/they don't understand me | 23 | 10.8 |
| | Somewhat well | 41 | 19.2 |
| | Very well | 149 | 70.0 |
| **Being Bullied** | Rarely/never | 177 | 83.1 |
| | Some days | 32 | 15 |
| | Most days | 4 | 1.9 |
| **Bullying Others** | Rarely/never | 200 | 93.9 |
| | Some days | 13 | 6.1 |
| **Sexually Active** | No | 204 | 95.8 |
| | Yes | 9 | 4.2 |
| **Sex Frequency** | Rarely/never | 204 | 95.8 |
| | Some days | 8 | 3.8 |
| **Sexual Abuse** | No | 196 | 92 |
| | Yes | 17 | 8 |
| **Substance use** | | | |
| **Alcohol Use** | Rarely/never | 206 | 96.7 |
| | Some days | 6 | 2.8 |
| | Most days | 1 | 0.5 |
| **Cigarette Use** | Rarely/never | 210 | 98.6 |
| | Some days | 2 | 0.9 |
| | Most days | 1 | 0.5 |
| **Cannabis Use** | Rarely/never | 207 | 97.2 |
| | Some days | 6 | 2.8 |
| **Chronic Illness** | No | 182 | 85.4 |
| | Yes | 31 | 14.6 |
| **Involvement in Sport Activity** | Rarely/never | 108 | 50.7 |
| | Some days | 89 | 41.8 |
| | Most days | 16 | 7.5 |

## Academic performance

The respondents' converted aggregate English and Mathematics scores ranged between 44 and 72. The median score was 56.0, while the mean score was 57.0 (SD = 6.4). There were no gender differences in academic performances, mean score for males was 57.42(SD = 6.8) compared to 56.61 (SD = 5.9) for females (t = -0.941; $p$ = 0.348).

## Rosenberg Self-Esteem Scale

The mean score for the whole sample was 19.07 (SD = 3.5). Boys reported higher scores compared to girls [19.55 (SD = 3.6) vs 18.61 (SD = 3.3; t = -1.997; $p$ = 0.049].

## Prevalence of depressive disorders

Twelve-month prevalence for any depressive disorder was 12.2%, major depressive disorder was 11.3% and dysthymia was 1.4%. Twelve-month prevalence for suicidality was 8.9%.

## Correlates of depressive disorders

These are shown in Table 3. Female gender ($p$ = 0.049); mother's educational attainment ($p$ = 0.021); perceived family dysfunction ($p$ < 0.001); domestic violence ($p$ = 0.046); history of

**Table 3. Correlates of depressive disorders.**

| | | DEPRESSIVE DISORDER | | | | |
|---|---|---|---|---|---|---|
| | | **No** | **Yes** | $\chi^2$ | **df** | **$p$-value** |
| | | **n (%)** | **n (%)** | | | |
| **Gender** | Female | 91 (83.5) | 18 (16.5) | 3.865 | 1 | **0.049** |
| | Male | 96 (92.3%) | 8 (7.7) | | | |
| **Mother's Education** | No formal education | 2 (66.7) | 1 (33.3) | 11.538 | 4 | **0.021** |
| | Primary | 56 (91.8) | 5 (8.2) | | | |
| | Secondary | 64 (95.5) | 3 (4.5) | | | |
| | Tertiary | 56 (78.9) | 15 (21.1) | | | |
| **Domestic Violence** | Rarely/never | 171 (89.5) | 20 (10.5) | 6.175 | 2 | **0.046** |
| | Some days | 13 (76.5) | 4 (23.5) | | | |
| | Most days | 3 (60.0) | 2 (40.0) | | | |
| **Family Functioning** | Highly Dysfunctional Family | 3 (50.0) | 3 (50.0) | 20.434 | 2 | **0.000** |
| | Moderately Dysfunctional Family | 15 (62.5) | 9 (37.5) | | | |
| | Highly Functional Family | 169 (92.3) | 14 (7.7) | | | |
| **Recent Loss** | No | 137 (91.3) | 13 (8.7) | 5.930 | 1 | **0.015** |
| | Yes | 50 (79.4) | 13 (20.6) | | | |
| **Close Friends** | None | 14 (66.7) | 7 (33.3) | 9.814 | 3 | **0.020** |
| | One | 31 (88.6) | 4 (11.4) | | | |
| | Two | 50 (90.9) | 5 (9.1) | | | |
| | Three or more | 92 (90.2) | 10 (9.8) | | | |
| **Getting Along with Friends** | Often quarrel/they don't understand me | 18 (78.3) | 5 (21.7) | 22.999 | 2 | **0.000** |
| | Somewhat well | 28 (68.3) | 13 (31.7) | | | |
| | Very well | 141 (94.6) | 8 (5.4) | | | |
| **Sexual Abuse** | No | 177 (90.3) | 19 (9.7) | 14.468 | 1 | **0.000** |
| | Yes | 10 (58.8) | 7 (41.2) | | | |
| **Being Bullied** | Rarely/never | 162 (91.5) | 15 (8.5) | 13.740 | 2 | **0.001** |
| | Some days | 22 (68.8) | 10 (31.3) | | | |
| | Most days | 3 (75.0) | 1 (25.0) | | | |

sexual abuse ($p < 0.001$), being bullied ($p = 0.001$); experiencing a recent loss ($p = 0.015$); having fewer number of close friends ($p = 0.020$); and difficulty getting along with friends ($p < 0.001$) were all associated with depressive disorders. There was no association between age, parental psychopathology, sibship, single-parent household, polygamous family setting, use of psychoactive substances and chronic medical conditions, and depression.

### Relationship between academic performance and disorder type

The aggregate English and mathematics exam scores were standardized to a normative mean of 50 and a standard deviation of 10. The standardization was done by obtaining a z score, which was multiplied by 10 and added to 50. The standardized scores were then compared between respondents diagnosed with depressive disorders vs. unaffected peers. Respondents with depressive disorder had a mean score of 44.71 (SD = 7.40), which was lower than the mean score of 50.74 (SD = 10.11) for non-depressed peers (t = 2.929; df = 211; $p = 0.004$).

### Relationship between self-esteem and disorder type

Participants without depressive disorder had a mean score of 19.67 (SD = 3.181) compared to those with depressive disorder with a mean score of 16.42 (SD = 16.42), ($F = 9.134$, $df = 3;209$, $p < 0.001$).

### Independent predictors of depressive disorders

In the multivariate regression analyses (see Table 4), female gender (OR = 4.250; $p = 0.046$), moderate difficulty getting along with friends (OR = 7.502; $p = 0.004$), the experience of being occasionally bullied (OR = 7.960; $p = 0.004$), history of sexual abuse (OR = 8.055; $p = 0.010$) and moderate family dysfunction (OR = 4.934; $p = 0.023$) were associated with depression.

**Table 4. Multivariate logistic regression of independent correlates of depressive disorders.**

| | B | OR | $p$ value | 95% C.I. for OR | |
|---|---|---|---|---|---|
| | | | | Lower | Upper |
| **Gender** | | | | | |
| Female | 1.447 | 4.250 | **0.046** | 1.026 | 17.600 |
| Male | Reference | 1.00 | | | |
| **Getting Along with Friends** | | | | | |
| Quarrelling/misunderstanding | 0.114 | 1.121 | 0.901 | 0.186 | 6.749 |
| Somewhat well | 2.183 | 8.874 | **0.004** | 2.045 | 38.514 |
| Very well | Reference | 1.00 | | | |
| **Sexual Abuse** | | | | | |
| Yes | 2.086 | 8.055 | **0.010** | 1.965 | 32.251 |
| No | Reference | 1.00 | | | |
| **Being Bullied** | | | | | |
| Some days | 2.074 | 7.960 | **0.004** | 1.965 | 32.251 |
| Most days | 2.317 | 10.149 | 0.121 | 0.541 | 190.265 |
| Rarely/Never | Reference | 1.00 | | | |
| **Family Functioning** | | | | | |
| Highly Dysfunctional Family | 1.377 | 3.965 | 0.301 | 0.291 | 54.065 |
| Moderately Dysfunctional Family | 1.596 | 4.934 | **0.023** | 1.241 | 19.613 |
| Highly Functional Family | Reference | 1.00 | | | |

## Discussion

This study aimed to determine the prevalence and correlates of depressive disorders among senior students attending secondary schools in Abeokuta. The twelve-month prevalence of any depressive disorder was 12.2%, and major depressive episode was 11.3%.

Our study shows that depressive syndromes are a significant health burden in adolescents. The reported prevalence in this study is similar to that reported in other studies [1, 2, 22]. The apparent convergence of rates is remarkable, given that these studies used different diagnostic instruments. Nevertheless, these findings are higher than elsewhere in Africa [7, 36] and the West [50, 51]. Differing prevalence periods could explain the varied findings. In addition, the present study used a twelve-month window and found a rate closer to the lifetime rate reported in the US comorbidity survey [1]; lower rates tended to be reported by studies using narrower time frames for prevalence rates.

The prevalence of dysthymia reported in this study was 1.4%, and to the best of the authors' knowledge is the first estimate of the disorder among adolescents in Nigeria. The other available estimate on dysthymia comes from the Nigerian National Survey of Mental Health and Wellbeing (NSMHW) by Gureje et al., which found a twelve-month prevalence of 0.1% in the adult population [52]. Similar prevalence rate of dysthymia has been reported in Norway [8] and Uganda [7]. These findings may suggest that although depressive syndromes occur frequently among adolescents, only a small proportion of episodes tend to run prolonged episodes.

In the bivariate analysis, gender and mother's education were the two sociodemographic variables significantly associated with depressive disorders in this study. This significant excess of females among adolescents having a depressive disorder has been replicated in many studies, both local and foreign [31, 53]. Gender difference in adolescence generally emerges in middle adolescence, typically by age 13 [53]. This has often led to suggestions that the female excess may be linked to pubertal changes in girls, specifically the achievement of Tanner stage III or IV [53]. However, the changes in body morphology associated with puberty and their resultant psychosocial effects on social interactions and self-perception are insufficient to explain the female adolescent depression excess and underlying changes in androgen and estrogen levels may play a significant mediating role [54].

In this study, depressive disorders appeared to be more prevalent in adolescents with parents with lower educational attainment, although this difference was only significant for mothers' educational attainment. Findings from elsewhere mostly report the reverse to be the case, i.e., adolescents with depression tend to have less educated parents [1, 7, 22]. While it may be true that for some unknown reasons, depressive disorders occur more frequently in adolescents with better-educated parents in this population, the method of collating the information should be put in perspective. Adolescents' self-report on parental educational attainment might not be a reliable way of determining the parents' highest level of education, so the finding should be interpreted with caution. Parental employment status, as reported by the adolescents, was the other approximate indicator of socioeconomic status included in this study, the analysis of which showed that depression was more prevalent in respondents with unemployed parents. However, this difference was not found to be significant. Yet it appears more in keeping with the frequent associations observed between adolescent depression and low socioeconomic status in other studies [55].

The presence of a depressive disorder was significantly associated with the reported occurrence of domestic violence and perceived family dysfunction in this study, similar to findings elsewhere [7, 22, 29, 36]. Persistent family disagreement through early adolescence increases the general level of low mood and depressive symptoms over time, and it is this rising level of

non-clinical negative mood and thoughts that is associated with the onset of later clinical depression in older adolescents [30, 56]. Conversely, it may also be true that depressive symptoms in adolescents precipitate conflicts in an otherwise normally functioning family [24, 57].

Reported traumatic experiences such as a recent loss within the last year, a history of sexual abuse, and being bullied were seen to be significantly associated with having a depressive disorder in this study, which is similar to other studies [29, 58]. Classic Freudian theory which explains depression as aggression displaced from an external hostile object and turned inwards against the self, provides a psychodynamic framework for understanding this association. However, there is more evidence in the research base to support the cognitive formulation, which proposes that early adverse experiences could result in an enduring triad of negative cognitions about the self, the world, and the future, which then become embedded as a latent negative schema and is activated by subsequent events [59, 60]. The significant association between depression and the reported experience of being bullied may be explained by Seligman's learned helplessness theory, which proposes that frequent exposure to uncontrollable and unpredictable events leads to an enduring loss of adaptive behaviors, eventually resulting in permanent deficits in cognitive and emotional processes [61].

Significant associations were also identified with impairments in peer relations in this study. Adolescents diagnosed with depressive disorder tended to report having fewer or no close friends at all and appeared to be experiencing difficulties getting along with friends. This agrees with findings by Field et al., who showed that depressed adolescents had less optimal peer relationships, fewer friends, and were less popular than unaffected peers [62]. Although interpersonal difficulties appear more to be a consequence rather than an antecedent of depression in adolescents, there is evidence that heightened sociotropy (an increased need or desire for peer approval) in the adolescent may render them more vulnerable to depression in the context of ongoing relational dysfunction with friends [62].

Regarding self-esteem, those with depressive disorders reported lower self-esteem compared to those who did not have depression, similar to other studies [21, 22, 63]. Two possible models explain the observed strong link between depression and reduced self-esteem. The vulnerability model hypothesizes that low self-esteem is a risk factor for depression, whereas the scar model hypothesizes that low self-esteem is an outcome, not a cause, of depression. The direction of causality appears to have been resolved in favor of the vulnerability model by longitudinal studies utilizing cross-lagged regression analyses, which indicated that low self-esteem predicted subsequent levels of depression and not vice versa [63].

Depressive disorders were significantly associated with reduced academic performance in this study, which is similar to findings by Hysenbegasi and colleagues [64]. Depression is associated with reduced volition, impaired concentration, and a general loss of interest in day-to-day tasks. Beyond potentially causing school absenteeism, affected adolescents might not fully engage in academic activities even when they attend school. Furthermore, intrinsic cognitive deficits are a recognized neuropsychological endophenotype of depression and may further limit academic performance even in the presence of sufficient engagement with school work [65].

In the multivariate regression analysis, female gender, the experience of bullying, difficulty getting along with friends, history of sexual abuse, lower self-esteem, and perceived family dysfunction were found to be independent predictors of depressive disorders, which is similar to what has been reported in other studies from around the world [1, 21, 22]. Traumatic childhood experiences, in particular, are well-recognized as strong predictors for the subsequent onset of emotional disorders and even other psychiatric disorders [66, 67].

The regression model could only explain 50.2% of the variance in depressive illnesses in this study. This shows that a significant percentage of the variation in depressive disorders

among adolescents may be explained by factors not considered in the current model, such as biological vulnerability. According to a meta-analysis of the genetic epidemiology of depression, the heritability was around 37% [68]. Such additive genetic factors are thought to moderate the risk of the onset of major depression in part by altering the sensitivity of individuals to some of the depression-inducing psychosocial stressors identified in the present study.

This study comes with some limitations. Firstly, due to the study's cross-sectional nature, causal inferences cannot be made. Secondly, the study was limited to a single city in the Southwest part of the country and may not be generalizable to adolescents in Nigeria. Thirdly, some of the questionnaires were self-reported and may be subject to response bias and social desirability bias, which may affect study validity. Fourth, the small sample size may limit the study's power to detect statistically significant associations. Nevertheless, the final sample size was more than the calculated minimum sample size needed to detect a difference. The study also has strengths: this study went beyond the scope of previous work done in Nigeria on adolescent depression to investigate additional possible correlates, such as parental educational attainment. This study was also the first to report the prevalence of dysthymia among the adolescent population in Nigeria.

In conclusion, this study demonstrated that the prevalence of depressive illnesses in our environment is at par with reports from other parts of Africa and the rest of the world. The study also emphasized characteristics associated with depression, including being bullied, having a history of sexual abuse, having low self-esteem, and family dysfunctionality. Studies with larger adolescent samples that combine structured diagnostic interviews with self-report depression instruments in a two-stage design may be able to find other significant relationships that were missed in the current study. Our study's findings underscore the importance of implementing depression screening initiatives for adolescents in secondary schools in Nigeria. Furthermore, we recommend providing training for guidance counselors to effectively identify and address depression in students exhibiting declining academic performance. Additionally, the prevention of bullying is also an important strategy that should be implemented to curb the incidence of depression among adolescents in secondary schools in Nigeria.

## Supporting information

**S1 Data.**
(SAV)

## Author Contributions

**Conceptualization:** Chukwuemeka Nzeakah, Lucky Onofa.

**Data curation:** Chukwuemeka Nzeakah, Temitope Ogundare.

**Formal analysis:** Chukwuemeka Nzeakah, Lucky Onofa.

**Funding acquisition:** Chukwuemeka Nzeakah, Oluwagbemiga DadeMatthews.

**Investigation:** Chukwuemeka Nzeakah, Lucky Onofa.

**Methodology:** Adefunke DadeMatthews, Chukwuemeka Nzeakah, Lucky Onofa, Oluwagbemiga DadeMatthews, Temitope Ogundare.

**Project administration:** Adefunke DadeMatthews, Chukwuemeka Nzeakah, Lucky Onofa.

**Resources:** Adefunke DadeMatthews, Chukwuemeka Nzeakah, Oluwagbemiga DadeMatthews.

**Software:** Adefunke DadeMatthews, Chukwuemeka Nzeakah.

**Supervision:** Adefunke DadeMatthews, Lucky Onofa.

**Validation:** Adefunke DadeMatthews, Chukwuemeka Nzeakah, Oluwagbemiga DadeMatthews, Temitope Ogundare.

**Visualization:** Adefunke DadeMatthews, Chukwuemeka Nzeakah, Oluwagbemiga DadeMatthews, Temitope Ogundare.

**Writing – original draft:** Chukwuemeka Nzeakah.

**Writing – review & editing:** Adefunke DadeMatthews, Lucky Onofa, Oluwagbemiga DadeMatthews, Temitope Ogundare.

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
