## [Decision Letter · Decision Letter 0]

13 Dec 2023

PONE-D-23-33725Teenage Blues: predictors of depression among adolescents in NigeriaPLOS ONE

Dear Dr. DadeMatthews,

Thank you for submitting your manuscript to PLOS ONE. After careful consideration, we feel that it has merit but does not fully meet PLOS ONE’s publication criteria as it currently stands. Therefore, we invite you to submit a revised version of the manuscript that addresses the points raised during the review process.

**Please pay close attention to remarks left by Reviewer 3. **==============================

We look forward to receiving your revised manuscript.

Kind regards,

Lakshit Jain, MD

Academic Editor

PLOS ONE

Reviewers' comments:

Reviewer's Responses to Questions

**Comments to the Author**

1. Is the manuscript technically sound, and do the data support the conclusions?

Reviewer #1: Yes

Reviewer #2: Yes

Reviewer #3: Yes

Reviewer #4: Partly

Reviewer #5: Yes

Reviewer #6: Yes

Reviewer #7: Yes

2. Has the statistical analysis been performed appropriately and rigorously? 

Reviewer #1: Yes

Reviewer #2: I Don't Know

Reviewer #3: No

Reviewer #4: N/A

Reviewer #5: Yes

Reviewer #6: I Don't Know

Reviewer #7: Yes

3. Have the authors made all data underlying the findings in their manuscript fully available?

Reviewer #1: Yes

Reviewer #2: Yes

Reviewer #3: Yes

Reviewer #4: Yes

Reviewer #5: Yes

Reviewer #6: Yes

Reviewer #7: No

4. Is the manuscript presented in an intelligible fashion and written in standard English?

Reviewer #1: Yes

Reviewer #2: Yes

Reviewer #3: Yes

Reviewer #4: Yes

Reviewer #5: Yes

Reviewer #6: Yes

Reviewer #7: Yes

5. Review Comments to the Author

Reviewer #1: The identification of independent predictors like female gender, bullying, sexual abuse, family dysfunction, and difficulty with friends for depressive disorders is strongly supported by multivariate regression analyses. Such analyses are standard for isolating predictive variables in observational studies. While the predictors align with known risk factors in the literature globally, such as the relationship between bullying and depression, the absence of detailed methodology questions whether the findings can be directly applied to all adolescents in Nigeria. Are there any studies in Nigeria backing up this finding? Also addressing the confounding factors, for example, the study could claim that experiencing bullying is associated with a higher prevalence of depression, but not necessarily that bullying causes depression without further longitudinal analysis. Good study overall.

Reviewer #2: The manuscript is based on impressive empirical evidence and makes an original contribution to the current available knowledge and reaffirms the existing knowledge of depression in adolescents and school going children. Professional use of English language is at par. It is interesting to note that mother’s education level is playing a role in developing depression in child’s future. The discussion section is little bit long and could be shortened to improve readability. Author has already mentioned the limitations of the study which is useful for future research conduction. Overall, well written manuscript.

Reviewer #3: I have read with great interest the article titled "Teenage Blues: predictors of depression among adolescents in Nigeria". The article presents an insight into the prevalence and correlates of depression in the Africa which have not been well studied. The article is well-written and presented in an organized manner. However, some of the statistical analysis are inaccurate and needs to re-checked and corrected.

1. The sample size is mentioned as 214 but most of the sample size in the tables total upto 213. Please explain the missing sample or correct the sample size number.

2. Please check the variable percentages in the paragraph "child-related psychosocial factors". Respondents who lost a close relative is mentioned as 28.2% while the table mentions it as 29.6%. Similar errors seen with other variables mentioned as well.

3. Table 3: I guess chi-square was used for the correlation. Please note chi square can be used only for variables with 2 sub-groups. For variables with more than 2 sub-groups (religious participation, father employment, father and mother education, caregiver, sibship, domestic violence, family functioning and others). Please use correlation tests like spearman/pearsons

4. It was mentioned that Pearson's correlation tests were used for variables. However no such results were reported. Please remove if tests not done or reported. Although, it would value to the study if these tests were done and reported.

5. It is mentioned as "Relationship between Academic Performance and Disorder Type" and "Relationship between Self-Esteem and Disorder Type". However, the tests reported in these are t-tests which are a test of differences and not 'relationship or association'. Please change accordingly.

6. It would be better if the tests of differences (t-tests, ANOVA) are presented in a table with p values. If space is a constraint, Table 3 and 4 can be modified to only present data for variables with a significant p value

Reviewer #4: Reviewer’s Feedback

Dear Editor,

Thank you for the opportunity to review the manuscript with the title “Teenage Blues: predictors of depression among adolescents in Nigeria”. The manuscript is an interesting read and provides knowledge about the burden depression among adolescent students in Abeokuta, Nigeria. However, there are some drawbacks.

Introduction

Line 96: Full stop is missing at the end of psychosocial functioning.

Methodology

Sampling and sample size: The criteria for selecting the 5 schools out the 39 public schools are not clear, noting that some of these public schools may be relatively different in some socio-demographic/economic characteristics based on where they are sited. Some schools with specific characteristics within this sub population may have been missed by just picking few schools randomly from these numbers. Again, private schools which constitute a significant proportion of this subgroup of adolescents in secondary schools with possibly different characteristics were not included. These factors may significantly affect the generalizability of the results. These I think are major setbacks, though some were mentioned.

Data Analysis: The later version of SPSS, version 23 and above are referred to as IBM SPSS statistics version (.......) as against Statistical Package for Social Sciences (SPSS) for Windows.

Results

Table 3: Correlates of Depressive Disorders

To improve the meaningfulness of the data, those variables that are less than 5 can be merged with closet options, more importantly for: mother's education, domestic violence, family functioning, sex frequency, being bullied. For example, for sexual frequency, most days and Some days can be merged together, FAMILY FUNCTIONING (Highly Dysfunctional and Moderately Dysfunctional Family can be merged together) etc.

Again, the χ2 and df can be removed from text, they do not add any additional information.

Similar adjustment can be made for similar variables in the multivariate regression analyses in Table 4.

Discussion

Line 265: “The reported prevalence in this study is that reported in other studies” This is not clear. Please, recast.

Reviewer #5: Thank you for the opportunity to review this manuscript that describes a descriptive study looking at prevalence of depressive disorders among a sample of adolescents in Nigeria.

The manuscript is well written and the methods are solid with good conclusions.

The major concern I have is the question on what this study adds to the literature. The authors themselves refer to previous studies among Nigerian adolescents. In the conclusion do mention looking at additional possible correlates, could this be expanded more in the background section ?

A few other thoughts:

- In the introduction section, in the lines 63-72: Since this is an international sample of adolescents, I would recommend including if the prevalence/risk data was from US samples or international to provide more context.

- Well written introduction section; gives a good background and establishes the rationale for the study well. One change that would make the flow better is moving the paragraph starting with ‘parental depressive symptoms…’ (line 114-120) to before the previous paragraph. That way all the risk factors are together and then the previous prevalence studies in the Nigerian population can be described.

- The methods are well described. In the first paragraph, it would be helpful to add at least a line or two describing the larger study that would help the authors understand the context for the data and the choice of schools.

- line 193, what is SSS 3 ?

- line 203, 28.2 % of respondents had a close relative in the last year. This seems pretty high. What do the authors attribute this to. Or how is ‘close relative’ defined. Also this number is 29.6 in Table 2. Why is there a discrepancy ? There are other discrepancies between the text and the table as well. Sexual abuse 14 vs 17 ? Substance use numbers seem different as well.

-In the abstract, conclusion section, I would revise the sentence to state ‘depressive syndrome area significant health burden among Nigerian adolescents’ to avoid the conclusion being too general.

Reviewer #6: The manuscript by Nzeakah et al. studies the important topic of depression among adolescents.

Strengths of the paper:

-The title is appropriate for the text's content.

-The introduction to the article is well laid out.

-The methods section clearly describes the study design, subject selection, sample size, etc.

-The discussion and conclusion further elaborate on the study objectives and limitations.

Some recommendations are:

-In the Discussion section (for example, lines 267 and 270), the authors reference data that is more than 10 years old to compare their study findings. I will recommend using more recent studies (e.g., PMID: 36272761, PMID: 34663534).

Reviewer #7: In this paper, the authors discuss the prevalence and correlates of depression in specific schools in Nigeria. I have the following comments/feedback for the authors:

The authors present a comprehensive review of the epidemiology of depression and its correlates among adolescents. However, the specific context for Nigeria appears to be missing. The paper can benefit from additional context and information about the differences between global and local trends in depression epidemiology and risk factors specific to Nigeria. This would also make the knowledge gap clear. The authors need to write at least one paragraph explaining. Why is this study important, and what does it add to the extensive literature on this topic?

Why was the time frame for depressive disorders in MINI-KID changed to 12 months? Does this have any impact on the validity of the instrument?

Were the selected schools representative of the schools in the region? In addition to informed consent from parents was ascent from the participants taken?

There are several inconsistencies in the formatting of the tables. For example, In Table 3, the actual counts (n) for each variable's "Yes" category are not provided. Also, please add confidence intervals where relevant.

The result section generally lacks a discussion about the findings' clinical significance or practical implications.

Can the authors provide some context or explanation for the score ranges selected for RSES?

The discussion section would further benefit from a statement about how these findings may contribute to adolescent mental health in Nigeria.

6. PLOS authors have the option to publish the peer review history of their article (what does this mean?). If published, this will include your full peer review and any attached files.

Reviewer #1: No

Reviewer #2: No

Reviewer #3: **Yes: **Nikhil Tondehal

Reviewer #4: No

Reviewer #5: No

Reviewer #6: No

Reviewer #7: No

---

## [Author Response · Author response to Decision Letter 0]

26 Dec 2023

Dear Editor,

Rebuttal Letter

The following are points raised in response to reviewers and details of revisions made. The reviewer’s comments are in black ink while the authors’ comments are in red ink.

Reviewer #1: The identification of independent predictors like female gender, bullying, sexual abuse, family dysfunction, and difficulty with friends for depressive disorders is strongly supported by multivariate regression analyses. Such analyses are standard for isolating predictive variables in observational studies. While the predictors align with known risk factors in the literature globally, such as the relationship between bullying and depression, the absence of detailed methodology questions whether the findings can be directly applied to all adolescents in Nigeria. Are there any studies in Nigeria backing up this finding? Also addressing the confounding factors, for example, the study could claim that experiencing bullying is associated with a higher prevalence of depression, but not necessarily that bullying causes depression without further longitudinal analysis. Good study overall.

Response: 

1. The study is representative of the population of adolescents in Abeokuta, given the our sampling technique. However, the results cannot be generalized to the whole adolescent in Nigeria, this was explicitly stated in the limitation section 

2. We were clear in the discussion to state association and not causation between the predictor variables and depression. Nowhere in our manuscript did we claim causation. 

Reviewer #2: The manuscript is based on impressive empirical evidence and makes an original contribution to the current available knowledge and reaffirms the existing knowledge of depression in adolescents and school going children. Professional use of English language is at par. It is interesting to note that mother’s education level is playing a role in developing depression in child’s future. The discussion section is little bit long and could be shortened to improve readability. Author has already mentioned the limitations of the study which is useful for future research conduction. Overall, well written manuscript.

Response: Thank you for your comment. Given the nature of the study and our findings, we think the discussion length is appropriate to discuss the findings, their clinical implications, and situate them in the context of existing literature. 

Reviewer #3: I have read with great interest the article titled "Teenage Blues: predictors of depression among adolescents in Nigeria". The article presents an insight into the prevalence and correlates of depression in the Africa which have not been well studied. The article is well-written and presented in an organized manner. However, some of the statistical analysis are inaccurate and needs to re-checked and corrected.

1. The sample size is mentioned as 214 but most of the sample size in the tables total upto 213. Please explain the missing sample or correct the sample size number.

Response: Thank you for catching the error! We have corrected this. (see line 204)

2. Please check the variable percentages in the paragraph "child-related psychosocial factors". Respondents who lost a close relative is mentioned as 28.2% while the table mentions it as 29.6%. Similar errors seen with other variables mentioned as well.

Response: Thank you for catching this error! This has been corrected. (See line 213-218)

3. Table 3: I guess chi-square was used for the correlation. Please note chi square can be used only for variables with 2 sub-groups. For variables with more than 2 sub-groups (religious participation, father employment, father and mother education, caregiver, sibship, domestic violence, family functioning and others). Please use correlation tests like spearman/pearsons

Response: Thank you for your comment. Chi-Square is also known as Pearson’s Chi-Square. It is commonly taught using a 2 x 2 table. However, Chi-square can be applied to 2 x 3 or more cross-tabulations, up to 5 x 5 tables. [see: https://www.bmj.com/about-bmj/resources-readers/publications/statistics-square-one/8-chi-squared-tests;
https://www.statology.org/chi-square-test-of-independence-calculator/ ]

4. It was mentioned that Pearson's correlation tests were used for variables. However, no such results were reported. Please remove if tests not done or reported. Although, it would value to the study if these tests were done and reported.

Response: Thank you for catching this error! We use Pearson correlation coefficient to determine association between self-esteem and academic performance, and family functioning and academic performance. However, we did not report these results. We have deleted that section from the methods section (Line 184-186). 

5. It is mentioned as "Relationship between Academic Performance and Disorder Type" and "Relationship between Self-Esteem and Disorder Type". However, the tests reported in these are t-tests which are a test of differences and not 'relationship or association'. Please change accordingly.

Response: Thank you for your comments. From our extensive knowledge and experience with academic writing and review of literature, it is correct to use the word ‘relationship’ when testing statistical hypothesis. A relationship between two or more variables can be an association, causation, an effect modifier, or confounding. What is most important is how these relationships are tested and reported. In this study, we clearly reported that the relationship between those variables were associations and not causations or other relationships. [See: https://www.sagepub.com/sites/default/files/upm-binaries/33663_Chapter4.pdf]

6. It would be better if the tests of differences (t-tests, ANOVA) are presented in a table with p values. If space is a constraint, Table 3 and 4 can be modified to only present data for variables with a significant p value

Response: Thank you for your suggestions. We considered this but decided that it would not be a good use of space. The t-test was used to measure the association between academic performance and depression. If we were to present this as a table, it would only have one row. Similarly, the ANOVA was used to test the relationship between self-esteem and depression. If we made this into a table, it will only have 1 row. We do not think this will be a good use of space. 

We have effected your suggestion of modifying our tables 3 and 4 to show only values that were statistically significant. 

Reviewer #4: Reviewer’s Feedback

Dear Editor,

Thank you for the opportunity to review the manuscript with the title “Teenage Blues: predictors of depression among adolescents in Nigeria”. The manuscript is an interesting read and provides knowledge about the burden depression among adolescent students in Abeokuta, Nigeria. However, there are some drawbacks.

Introduction

Line 96: Full stop is missing at the end of psychosocial functioning.

Response: Thank you for catching the error! It has been corrected. (see line 96)

Methodology

Sampling and sample size: The criteria for selecting the 5 schools out the 39 public schools are not clear, noting that some of these public schools may be relatively different in some socio-demographic/economic characteristics based on where they are sited. Some schools with specific characteristics within this sub population may have been missed by just picking few schools randomly from these numbers. Again, private schools which constitute a significant proportion of this subgroup of adolescents in secondary schools with possibly different characteristics were not included. These factors may significantly affect the generalizability of the results. These I think are major setbacks, though some were mentioned.

Response: Thank you for your comment. The role of sampling is to ensure that the sample size is representative of the population. In this study, we used a probability sampling technique, which gives each school an equal chance of being selected and reduces sampling bias. Our probability sampling was in 2 stages: one, in selecting the schools, divided into public and private. We did this because we realize that public and private schools may differ in sociodemographic variables, and distribution of our outcome variables. 

Secondly, within the schools selected, we used proportional systematic random sampling to select the students who would participate. From schools and classes with larger number of students, we selected larger number of students to participate. 

When the word ‘random’ is used in statistical sampling, it does not mean ‘on a whim’. It refers to probabilistic sampling method to ensure that every of the 39 schools had equal chance of being selected. 

In a separate paper, we described the methodology in full. 

Data Analysis: The later version of SPSS, version 23 and above are referred to as IBM SPSS statistics version (.......) as against Statistical Package for Social Sciences (SPSS) for Windows.

Response: Thank you for the correction, it has been revised accordingly (see line 179)

Results

Table 3: Correlates of Depressive Disorders

To improve the meaningfulness of the data, those variables that are less than 5 can be merged with closet options, more importantly for: mother's education, domestic violence, family functioning, sex frequency, being bullied. For example, for sexual frequency, most days and Some days can be merged together, FAMILY FUNCTIONING (Highly Dysfunctional and Moderately Dysfunctional Family can be merged together) etc.

Again, the χ2 and df can be removed from text, they do not add any additional information.

Similar adjustment can be made for similar variables in the multivariate regression analyses in Table 4.

Response: Thank you for your suggestions. Data presented in the tables are to give a snapshot of the results, and we wanted to make sure that it was easy for readers to be able to have this information. 

Also, it is standard practice to report measures of association when reporting results in scientific literature. Given that not all results can be presented in tabular form given constraints of space, it is important that we mention these measures of associations while reporting the results in the text. 

(See adjustments in line 236-241)

Discussion

Line 265: “The reported prevalence in this study is that reported in other studies” This is not clear. Please, recast.

Response: Thank you for catching the error, it has been corrected. (see line 275)

Reviewer #5: Thank you for the opportunity to review this manuscript that describes a descriptive study looking at prevalence of depressive disorders among a sample of adolescents in Nigeria.

The manuscript is well written and the methods are solid with good conclusions.

The major concern I have is the question on what this study adds to the literature. The authors themselves refer to previous studies among Nigerian adolescents. In the conclusion do mention looking at additional possible correlates, could this be expanded more in the background section ?

Response: Thank you for your insightful comment. As articulated in our introduction section, our primary objective was to contribute to the relatively sparse body of studies conducted in Nigeria. Despite the similarity of our results to existing research, it is essential to emphasize that this congruence does not diminish the significance of our study's contribution to the literature.

Moreover, our research stands out as the pioneer investigation into the prevalence of depressive disorders beyond major depressive disorder in Nigeria. Previous studies in the region have predominantly focused on major depressive disorders. By broadening the scope, we have provided a more comprehensive understanding of depressive disorders in the local context.

Adhering to the fundamental principle of scientific reproducibility, the consistency of our findings with other limited studies in Nigeria adds robustness to the identified risk factors. This convergence further instills confidence in the validity of our results, thereby offering valuable insights for the development of clinical and epidemiological interventions.

Additionally, the strength of our study lies in the utilization of a representative sampling technique, coupled with a notably larger sample size compared to previous studies in Nigeria. This enhances the reliability and generalizability of our findings, contributing to the overall advancement of knowledge in this field.

A few other thoughts:

- In the introduction section, in the lines 63-72: Since this is an international sample of adolescents, I would recommend including if the prevalence/risk data was from US samples or international to provide more context.

Response: Thank you for your comment. If you look at the first paragraph, we referenced not only US samples but studies in Nigeria. 

- Well written introduction section; gives a good background and establishes the rationale for the study well. One change that would make the flow better is moving the paragraph starting with ‘parental depressive symptoms…’ (line 114-120) to before the previous paragraph. That way all the risk factors are together and then the previous prevalence studies in the Nigerian population can be described.

Response: Thank you for your suggestion! We have revised accordingly. (see line 100-106)

- The methods are well described. In the first paragraph, it would be helpful to add at least a line or two describing the larger study that would help the authors understand the context for the data and the choice of schools.

Response: Thank you for your suggestion. We have revised accordingly. (see line 134-138)

- line 193, what is SSS 3 ?

Response: corrected [senior secondary school class 3 students] – line 202

- line 203, 28.2 % of respondents had a close relative in the last year. This seems pretty high. What do the authors attribute this to. Or how is ‘close relative’ defined. Also this number is 29.6 in Table 2. Why is there a discrepancy ? There are other discrepancies between the text and the table as well. Sexual abuse 14 vs 17 ? Substance use numbers seem different as well.

Response: Thank you for spotting the discrepancies! We have corrected them (see line 213-218)

Close relative includes 1st and 2nd degree relatives. Given the nature of Nigerian culture, the extended family is a very integral part of the family structure. We do not think the number is particularly high. The life expectancy in Nigeria is about 55 years.

-In the abstract, conclusion section, I would revise the sentence to state ‘depressive syndrome area significant health burden among Nigerian adolescents’ to avoid the conclusion being too general.

Response: Thank you for the suggestion, we have revised the section accordingly

Reviewer #6: The manuscript by Nzeakah et al. studies the important topic of depression among adolescents.

Strengths of the paper:

-The title is appropriate for the text's content.

-The introduction to the article is well laid out.

-The methods section clearly describes the study design, subject selection, sample size, etc.

-The discussion and conclusion further elaborate on the study objectives and limitations.

Some recommendations are:

-In the Discussion section (for example, lines 267 and 270), the authors reference data that is more than 10 years old to compare their study findings. I will recommend using more recent studies (e.g., PMID: 36272761, PMID: 34663534).

Response: Thank you for your suggestion. However, our study has over 68 references, most of which are recent. 

Reviewer #7: In this paper, the authors discuss the prevalence and correlates of depression in specific schools in Nigeria. I have the following comments/feedback for the authors:

The authors present a comprehensive review of the epidemiology of depression and its correlates among adolescents. However, the specific context for Nigeria appears to be missing. The paper can benefit from additional context and information about the differences between global and local trends in depression epidemiology and risk factors specific to Nigeria. This would also make the knowledge gap clear. The authors need to write at least one paragraph explaining. Why is this study important, and what does it add to the extensive literature on this topic?

Response: Thank you for your comment. In our introduction section, we meticulously cited existing literature in Nigeria to establish the context. We also provided our rationale: driven by the observation of limited studies, typically confined to one major town in the country. Notably, our study stands out as the first to investigate depressive disorders beyond major depressive disorder in this context.

Why was the time frame for depressive disorders in MINI-KID changed to 12 months? Does this have any impact on the validity of the instrument?

Response: The time frame was changed to capture the 12-month prevalence of depressive disorders. No, this does not have any impact on the validity, as the diagnostic criteria was left unchanged.

Were the selected schools representative of the schools in the region? In addition to informed consent from parents was ascent from the participants taken?

Response: Yes, we used a probability sampling method to ensure representative sample was collected. Yes, participants gave accent. We stated this in the methods section.

There are several inconsistencies in the formatting of the tables. For example, In Table 3, the actual counts (n) for each variable's "Yes" category are not provided. Also, please add confidence intervals where relevant.

Response: Thank you for your comment. Actually, we provided both the N and percentages in Table 3. We also provided confidence intervals in Table 4 which reported the OR from the multivariate regression analyses. 

The result section generally lacks a discussion about the findings' clinical significance or practical implications.

Response: The results section serves the purpose of reporting the findings, while the discussion section is dedicated to examining the study's results in terms of clinical significance and practical implications. We indeed conducted a thorough discussion in the relevant section.

Can the authors provide some context or explanation for the score ranges selected for RSES?

Response: in the methods section, we provided detailed information on the structure of the RSES and how it is scored. 

The discussion section would further benefit from a statement about how these findings may contribute to adolescent mental health in Nigeria.

Response: Thank you for your suggestion. We have implemented this in the manuscript (see line 386-391)

---

## [Decision Letter · Decision Letter 1]

9 Jan 2024

Teenage Blues: predictors of depression among adolescents in Nigeria

PONE-D-23-33725R1

Dear Dr. DadeMatthews,

We’re pleased to inform you that your manuscript has been judged scientifically suitable for publication and will be formally accepted for publication once it meets all outstanding technical requirements.

Kind regards,

Lakshit Jain, MD

Academic Editor

PLOS ONE

Additional Editor Comments (optional):

Reviewers' comments:

Reviewer's Responses to Questions

**Comments to the Author**

1. If the authors have adequately addressed your comments raised in a previous round of review and you feel that this manuscript is now acceptable for publication, you may indicate that here to bypass the “Comments to the Author” section, enter your conflict of interest statement in the “Confidential to Editor” section, and submit your "Accept" recommendation.

Reviewer #1: All comments have been addressed

Reviewer #2: All comments have been addressed

Reviewer #3: All comments have been addressed

Reviewer #5: All comments have been addressed

2. Is the manuscript technically sound, and do the data support the conclusions?

Reviewer #1: Yes

Reviewer #2: Yes

Reviewer #3: Yes

Reviewer #5: Yes

3. Has the statistical analysis been performed appropriately and rigorously? 

Reviewer #1: Yes

Reviewer #2: I Don't Know

Reviewer #3: Yes

Reviewer #5: Yes

4. Have the authors made all data underlying the findings in their manuscript fully available?

Reviewer #1: Yes

Reviewer #2: Yes

Reviewer #3: Yes

Reviewer #5: Yes

5. Is the manuscript presented in an intelligible fashion and written in standard English?

Reviewer #1: Yes

Reviewer #2: Yes

Reviewer #3: Yes

Reviewer #5: Yes

6. Review Comments to the Author

Reviewer #1: I commend the author for their swift and thorough response to the raised concerns. The revisions made have significantly improved the overall quality of the work, showcasing a keen attention to detail and a commitment to addressing the feedback constructively.

Reviewer #2: All the comments have been replied satisfactorily by the author. This is a good study and results are supportive to the claim.

Reviewer #3: All comments have been addressed. One small change needs to be made. Please report all p values that are '0.000' as '<0.001'. Article may be accepted after this change

Reviewer #5: The authors have addressed my comments satisfactorily.

Thank you for the opportunity to re-review this manuscript.

7. PLOS authors have the option to publish the peer review history of their article (what does this mean?). If published, this will include your full peer review and any attached files.

Reviewer #1: **Yes: **Aditi Sharma

Reviewer #2: No

Reviewer #3: No

Reviewer #5: No

---

## [Editor Report · Acceptance letter]

5 Apr 2024

PONE-D-23-33725R1 

PLOS ONE

Dear Dr. DadeMatthews, 

I'm pleased to inform you that your manuscript has been deemed suitable for publication in PLOS ONE. Congratulations! Your manuscript is now being handed over to our production team.

Kind regards, 

on behalf of

Dr. Lakshit Jain 

Academic Editor

PLOS ONE